# Ego-CNN: An Ego Network-based Representation of Graphs Detecting Critical Structures

## Abstract

While existing graph embedding models can generate useful embedding vectors for graph-related tasks, what valuable information can be jointly learned from a graph embedding model is less discussed. In this paper, we consider the possibility of detecting critical structures by a graph embedding model. We propose Ego-CNN to embed graphs, which works in a local-to-global manner to take advantages of CNNs that gradually expands the detectable local regions on the graph as the network depth increases. Critical structures can be detected if Ego-CNN is combined with a supervised task model. We show that Ego-CNN is (1) competitive to state-of-the-art graph embeddings models, (2) can work nicely with CNNs visualization techniques to show the detected structures, and (3) is efficient and can incorporate with scale-free priors, which commonly occurs in social network datasets, to further improve the training efficiency.

## 1 Introduction

A graph embedding algorithm converts graphs from structural representation to fixed-dimensional vectors. It is typically trained in a unsupervised manner for general learning tasks but recently, deep learning approaches such as Structure2vec (Dai et al. (2016)) and Diffusion Convolution Neural Network (Atwood & Towsley (2016)) are trained in a supervised manner and shown superior results against unsupervised approaches on many tasks such as node classification and graph classification.

While many algorithms perform well on graph-related tasks, what valuable information can be jointly learned from the graph embedding is less discussed. In this paper, we aim to develop a graph embedding model that jointly discovers the *critical structures*, i.e., partial graphs that are dominant to a learning task (e.g., graph classification) where the embedding is applied to. This helps people running the learning task to understand the reason behind the task results, and is particularly useful in certain domains such as the bioinformatics and social network analysis. For example, people in bioinformatics may wish to tell whether a protein is an enzyme or not and *why*. As shown in Figure 1, if there is model that can backtrack the most critical parts, it may help people understand that it is the circular structures (the "holes") that distinguish enzyme from other substances. New knowledge may then be discovered by investigating the functionality of these holes.

Positive       Negative

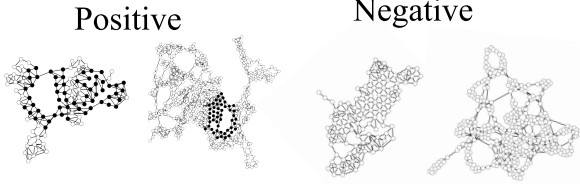

Figure 1: A model that detects critical structure (colored in black) for the graph classification task (enzymes classification in this case) can help people to understand why a protein is of a specified class or not.

Unfortunately, identifying critical structures is a challenging task. The first challenge is that critical structures are *task-specific*—the shape and location of critical structures may vary from task to task. This means that the graph embedding model needs to be learned together with the task model (e.g.,

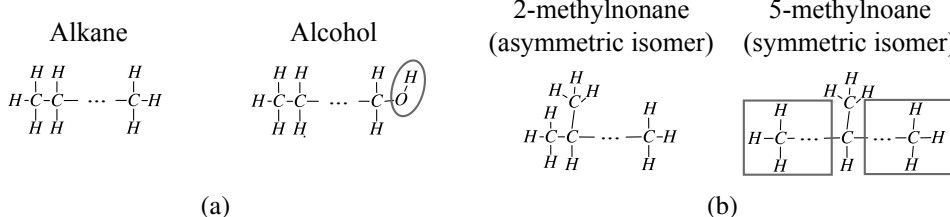

Figure 2: (a) OH function group is the critical structure to tell Alcohols from Alkanes. (b) The symmetry hydrocarbon group (at two sides of the methyl branch) is the critical structure to discriminate between symmetric and asymmetric isomer of methylnonane.

a classifier or regressor). Another challenge is that critical structures may be found at the *global-scale*, as shown in Figure 2. To discriminant Alcohols from Alkanes (Figure 2(a)), one can check if there exists an OH-base and if the OH-base is at the end of the compound. In this task, the critical structure (OH-base) is relative small and can be identified at the local-scale. But for the other task (Figure 2(b)), to identify if a methyl-nonane is symmetric or not, one must check the entire graph to know if the methyl is branched at the center position of the long carbon chain. In this task, the critical structure is the symmetric hydrocarbon at the two sides of the methyl branch, which can only be found at the global-scale. The second challenge is that finding out all matches of substructures in a graph is known as subgraph isomorphism and proven to be an NP-complete problem by Cook (1971).

To the best of our knowledge, there is no existing graph embedding algorithm that can identify critical structures up to the global-scale in an efficient way. Most existing approaches has only limited ability in identifying critical structures. For example, Structure2vec (Dai et al. (2016)) and Spectrum GCN (Defferrard et al. (2016)) can only find critical structures of very simple shape. Patchy-San (Niepert et al. (2016)) can only find critical structures within a small area around each node (i.e., at the local-scale). The only work that is able to identify critical structures at the global-scale is Spatial GCN (Bruna et al. (2013)), but it has the complexity $O(N^3)$, where $N$ is the number of nodes, in both time and space, therefore is inefficient for large graphs.

In this paper, we propose the *Ego-CNN* that embeds a graph into distributed (multi-layer), fixed-dimensional tensors. The Ego-CNN can be jointly learned with a supervised task model (e.g., fully-connected layers for graph classification) and help identify the task-specific critical structure at the global-scale. Some existing studies like DCNN (Atwood & Towsley (2016)) and Patchy-San (Niepert et al. (2016)) have borrowed the concept of CNN to embed graphs. The idea is to model the *filters/kernels* that scan through different parts of the graph (which we call the *neighborhood*s) to learn patterns most helpful to the learning task. Unlike images where a neighborhood can be clearly defined as the nearby pixels across channels, graphs have no standard notion of neighborhoods. Currently, the neighborhoods defined by above works have some limitations in that (1) a neighborhood does *not* represent a local region of the graph so the embedding model cannot take advantage of the location invariant patterns to speed up the learning process, or (2) a neighborhood represents a local region of the graph but cannot be recursively defined over the hidden layers, making the embedding model shallow and incapable of detecting critical structure at the global-scale. Different from most existing approaches, Ego-CNN defines a neighborhood at layer $l$ as the $(l + 1)$-hop ego network centered at a specific node.[1] This makes the Ego-CNN efficient since it learns the location invariant patterns. Also, a neighborhood at layer $l$ enlarges the receptive field of the neighborhood centered at the same node at layer $l - 1$. Ego-CNN can exploit the multi-layer architecture to detect the critical structure at the global-scale in the deeper layers. Furthermore, Ego-CNN works nicely with some common visualization techniques for CNNs (e.g., deconvolution proposed by Zeiler et al. (2011)) and can output the critical structures behind each prediction made by the trained task model. Our contributions are

- We propose the Ego-CNN and show that an graph embedding based on ego networks can perform as well as (if not better than) existing approaches. We conduct experiments on

---

[1]In a graph, an $l$-hop ego network centered at node $i$ is a subgraph consisting of the node $i$ and all its $l$-hop neighbors as well as the edges between these nodes.

many graph classification tasks and the results show that Ego-CNN can lead to the state-of-the-art performance.

- By letting a neighborhood (an ego network) at a deeper layer share the same centering node with a neighborhood at the preceding layer (which we call the *egocentric design*), Ego-CNN supports a multi-layer architecture with enlarged receptive fields at deeper layers, thus can learn the critical structures at the global-scale.

- Ego-CNN, which learns location invariant patterns recursively, is efficient. Each convolution takes only $O(Nkd)$ in both space and time for a filter to scan a graph with $N$ nodes. We also show that Ego-CNN can readily incorporate the scale-free prior, which commonly exists in large (social) graphs, to further improve the training efficiency.

To the best of our knowledge, Ego-CNN is the first embedding model that can efficiently detect task-dependent critical structures at the global-scale. We hope it helps practitioners better explain the learning results and discover new knowledge in the future.

## 2 RELATED WORK

We compare existing graph embedding models in Table 1 on the aspects of our goal, i.e. whether it jointly detects the global-scale critical structure and its computational complexity.

| Graph embedding model | Can identify critical structures (upto the global-scale)? | Efficient on large graph? | Time complexity to embed a graph $G = (V, E)$ |
|---|:---:|:---:|:---:|
| WL kernel (Shervashidze et al. (2011)) | | ✓ | $O(knL\lvert V\rvert)$ |
| DGK (Yanardag & Vishwanathan (2015)) | | ✓ | $O(kdn\lvert V\rvert)$ |
| Subgraph2vec (Narayanan et al. (2016)) | | ✓ | $O(kdn\lvert V\rvert)$ |
| MLG (Kondor & Pan (2016)) | | | $O(L\lvert V\rvert^5)$ |
| Structure2vec (Dai et al. (2016)) | | ✓ | $O(kd\lvert V\rvert)$ |
| Spatial GCN (Bruna et al. (2013)) | ✓ | | $O(dL\lvert V\rvert^3)$ |
| Spectrum GCN (Bruna et al. (2013); Defferrard et al. (2016); Kipf & Welling (2017)) | | ✓ | $O(dn_f\lvert E\rvert)$ |
| DCNN (Atwood & Towsley (2016)) | | | $O(hn_f\lvert V\rvert^2)$ |
| Patchy-San (Niepert et al. (2016)) | | ✓ | $O(k^2 n_f\lvert V\rvert)$ |
| Neural Fingerprint (Duvenaud et al. (2015)) | | ✓ | $O(kn_f\lvert V\rvert)$ |
| Ego-Convolution | ✓ | ✓ | $O(kdn_f L\lvert V\rvert)$ |

Table 1: A comparison of graph embedding models. Let $d$ be the embedding dimension, k be maximum node degree in $G$. Other variables such as $L, n_f, n$ are model dependent hyperparameters.

**Graph Kernels** All graph kernels have a common drawback in that the embeddings are generated in a unsupervised manner, which means critical structure cannot be jointly detected at the generation of embeddings. Here we introduce three state-of-the-art graph kernels. The Weisfeiler-Lehman kernel (Shervashidze et al. (2011)) grows the coverage of each node by collecting information from neighbors, which is conceptually similar to our method, but differs in that WL kernel collects only node labels, while our method collects the complete labeled neighborhood graphs from neighbors. Deep Graph Kernels (Yanardag & Vishwanathan (2015)) and Subgraph2vec (Narayanan et al. (2016)) were inspired by word2vec (Mikolov et al. (2013)) models that embed structures by predicting neighbors' structures. Multiscale Laplacian Graph Kernels (Kondor & Pan (2016)) can compare graphs at multiple scales by recursively comparing graphs based on the comparison of subgraphs. However, it's very inefficient in that comparing two graphs takes $O(L\lvert V\rvert^5)$, where $L$ represents the number of comparing scales.

**Graphical Models** Graph can also express the conditional dependence (edge) between random variables (node) in graphical models. models. Structure2vec (Dai et al. (2016)) introduced a novel layer that makes the optimization procedures of approximation inference directly trainable by SGD. It's efficient on large graphs with time complexity linear to number of nodes. However, it's weak at identifying critical structures since approximation inference makes too much simplification on structures. For example, mean-field approximation assumes variables are independent of each other. As a result, Structure2vec can only identify critical structures of very simple shape.

**Convolution-based Methods** Recently, many works are proposed by borrowing the concept of CNN to embed graphs. Figure 3 summarizes their definition of filters and neighborhoods.

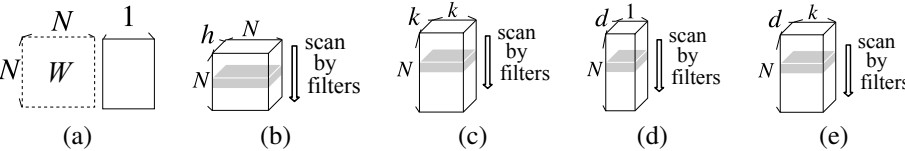

Figure 3: Filters and neighborhood in (a) Spatial GCN by Bruna et al. (2013). The filter $W$ is a sparse matrix, which is not aimed to scan for detecting local patterns, but to learn connectivity of clusters based on cluster features $X$. (b) DCNN by Atwood & Towsley (2016) that scans through the diffusion matrix of each node. (c) Patchy-San by Niepert et al. (2016) that scans adjacency matrix of local neighborhood of each node. (d) Neural Fingerprints by Duvenaud et al. (2015) that scans approximated neighborhood (specially, the summation of neighbors' embeddings) of each node. (e) Ego-Convolution scans egocentric designed neighborhoods of each node.

The Spatial Graph Convolutional Network (GCN) was proposed by Bruna et al. (2013). The design of Spatial GCN (Figure 3(a)) is very different from other convolution-based methods since its goal is to perform hierarchical clustering of nodes. A neighborhood is defined as a cluster. However, the filter is not aim to scan for local patterns but to learn the connectivity of all clusters. This means each filter is of size $O(N^2)$ if there're $N$ clusters. Also, a filter requires the global-scale information, i.e. the features of all clusters to train, so it's very inefficient on large graphs.

Hence, in the same paper, Bruna et al. (2013) proposed another version, the Spectrum GCN, to perform hierarchical clustering in the spectrum domain. Although the efficiency is improved and many recent work such as Defferrard et al. (2016); Kipf & Welling (2017) further improve the training efficiency, the major drawback is that spectrums is weak at identifying structures since only special families of graphs such as complete graphs or star-like trees can be precisely described by spectrums.

Diffusion Convolutional Neural Networks (Atwood & Towsley (2016)) embed graphs by detecting the diffusion. The neighborhood is defined as the diffusion, which are paths starting from a node to other nodes in $h$ hops. The diffusion can be represented by a diffusion matrix.[2] They use filters to scan through each node's diffusion matrix, so DCNN is possible to detect useful diffusion patterns. But it cannot detect critical structures since the diffusion patterns cannot precisely describe the location and the shape of structures. DCNN reported impressive results on node classification. But it's inefficient on graph classification tasks since their notion of neighborhood is at the global-scale, which takes $O(hN^2)$ to embed a graph with $N$ nodes. Also, their definition of neighborhood makes the embedding model shallow, which is only a single layer in the neural network.

Patchy-San (Niepert et al. (2016)) detects local neighborhoods by filters. The definition of local neighborhood is based on the adjacency matrix of the graph, which is defined as the local graphs formed by its $k$ nearest neighbors. The filter scans on the adjacency matrix of local neighborhoods. Since expressing graphs by adjacency matrices is not invariant under different vertex ordering, they proposed neighborhood normalization to generate similar adjacency matrices for isomorphic neighborhoods. Patchy-San can detect precise structures but it is inefficient to scan for large local neighborhood, which limits it from detecting global-scale critical structures. Also, the embedding model of Patchy-San is only a single layer, a question to ask is that is it possible to improve the efficiency by generalizing its definition of neighborhood into multiple layers? If the adjacency matrix of neighborhoods is generated (e.g. the similarity of all embedding vectors of all neighborhoods), the neighborhood at next layer can be defined as the $k$ most similar neighborhoods. However, this naive generalization don't work for two reasons. First, the concept of local neighborhood is lost since two neighborhoods that have similar embedding vectors are not necessary to be adjacent on the graph. Second, the structure of neighborhood is dynamic since the $k$ most similar neighborhoods changes during training. Thus, this generalization (based on adjacency matrix) does not give neighborhoods corresponding to local regions

Therefore, the idea of Patchy-San cannot be readily extended to multiple layers.

---

[2] A diffusion matrix $D$ is of size $h \times N$ for a graph of $N$ nodes and each element $D_{ij}$ represents if this node connects to node $v_j$ in $i$ hops.

To the best of our knowledge, the only work that scans on local neighborhoods and supports multiple layers is Neural Fingerprints (Duvenaud et al. (2015)). But the neighborhood at layer $l$ represents only the approximation (specially, the summation) of the $l$-hop neighbors. They use the approximation to avoid the vertex-ordering problem of adjacency matrix. However, at the same time, Neural Fingerprint loses the ability of detecting precise critical structures.

We know that CNN has two advantages: (1) detecting location invariant patterns and (2) overcoming the curse of dimensionality by multi-layer representations. To keep these advantages on graphs, we have to rethink the definition of neighborhood from scratch.

## 3 EGO-CNN

The reason why the idea of Patchy-San fails to generalize into multiple layers is that its definition of neighborhood (which is based on adjacency matrix) makes the composition of neighborhoods at deeper layer not *static* and may not correspond to local regions in the graph. Our main idea is to use the *egocentric design*, i.e. the neighborhood is defined on the *same node* as all the preceeding layers. The overview of our method is depicted in Figure 4.

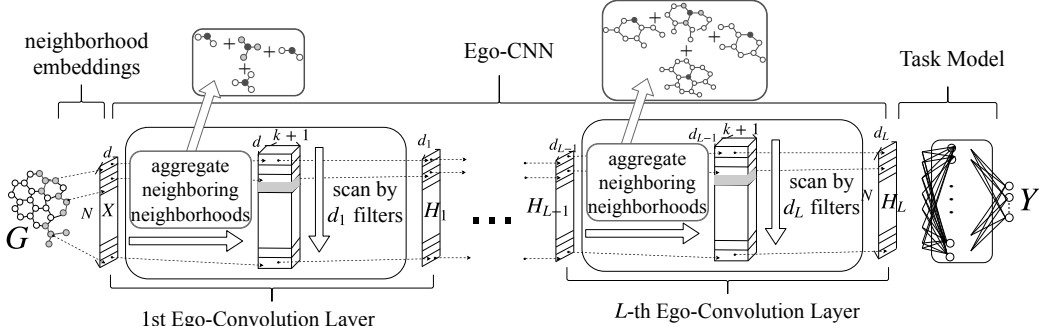

Figure 4: The model architecture of Ego-CNN. Our egocentric design defines neighborhood on the same node (dashed line indicates the neighborhoods centered at the same node). Neighborhoods are enlarged through Ego-Convolution by aggregating neighbors' neighborhoods.

Given a graph $G$ of $N$ nodes,

(1) Preprocess step: generate the neighborhood embeddings $h^{(0)} \in \mathbb{R}^{N \times d}$ and select the $k$ nearest neighbors $Nbr$ for all nodes

For each node $i$, we select its $k$ nearest neighbors $Nbr(i)$, take its neighborhood as the subgraph formed by including node $i$ itself and its $k$ neighbors, and then embed this neighborhood to a $d$-dim vector $h_i^{(0)} \in \mathbb{R}^d$. In general, any existing graph embedding models can be used to embed the neighborhood graphs, but we recommend to use Patchy-San since our Ego-CNN can be directly stacked on top of it.

(2) Ego-CNN: output embeddings of the enlarged neighborhoods

The Ego-CNN is composed of several Ego-Convolution layers. Our egocentric design is to define a neighborhood as its and its $k$ neighbors' neighborhoods at previous layer, which results in a larger neighborhood centered on the same node. Figure 5 shows how this definition leads to the enlarged neighborhoods. Suppose neighborhoods are originally 1-hop ego networks. Consider the bold black node in Figure 5, aggregating its neighbors' neighborhoods (centered in black nodes), its neighborhood is stretched out by 1-hop, becoming a 2-hop ego network.

Specifically, we give the formal definition of our Ego-Convolution. $A*B$ denotes $\sum_{i=1}^{M} \sum_{j=1}^{N} A_{ij} B_{ij}$ for two matrices $A, B$ of the same size. The $l$-th Ego-Convolution layer generates node $i$'s neighborhood embedding $h_i^{(l)} \in \mathbb{R}^{d_l}$ by $h_{ij}^{(l)} = \sigma \left( X_i^{(l)} * W_j^{(l)} + b_j^{(l)} \right), \forall j = 1, \cdots, d_l,$

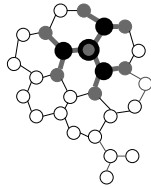

Figure 5: Our definition of neighborhoods is egocentric to a node (in bold black), which includes its(colored in black) and its neighbors' neighborhoods (centered in the black node).

where $\sigma$ is the activation function, $X_i^{(l)} = \begin{bmatrix} h_i^{(l-1)} \\ h_{n_1}^{(l-1)} \\ \vdots \\ h_{n_k}^{(l-1)} \end{bmatrix}$, $n_1, \cdots, n_k \in Nbr(i)$ are node $i$'s neighbors and $W_1^{(l)}, \cdots, W_{d_l}^{(l)} \in \mathbb{R}^{(k+1) \times d_{l-1}}$ are filters at $l$-th layer, and $b_1^{(l)}, \cdots, b_{d_l}^{(l)} \in \mathbb{R}$ are bias terms.

That is, $X_i^{(l)}$ is the concatenated matrix of its and its $k$ neighbors' neighborhood embeddings at previous layer. The ordering of its neighbors $n_1 \cdots n_k$ is determined using the neighborhood normalization proposed by Niepert et al. (2016) in the the preprocessing step, where the goal of using neighborhood normalization is to ensure two isomorphic neighborhoods to have the same concatenated matrix.

It is important to note that $X_i^{(l)}$ keeps the precise structure of node $i$'s neighborhood. As $h_{n_1}^{(l-1)} \cdots h_{n_k}^{(l-1)}$ preserve its neighbors' neighborhood graphs, and the relative information about how its $k$ neighbors' neighborhoods are connected is kept in $h_i^{(l-1)}$ and in the ordering of $k$ neighbors. Hence, by scanning through each $X_i^{(l)}$, the filters can capture precise neighborhood structures.

By contrast, previous works such as GCN(Kipf & Welling (2017)) and Neural Fingerprints(Duvenaud et al. (2015)) fail to learn precise neighborhood structure as they generate node embeddings equivalently by

$h_{ij}^{(l)} = \sigma \left( A_i^{(l)} * W_j^{(l)} + b_j^{(l)} \right)$, where $A_i^{(l)} = \left( c_0 h_i^{(l-1)} + c_1 h_{n_1}^{(l-1)} + \cdots + c_k h_{n_k}^{(l-1)} \right)$, $W_j^{(l)} \in \mathbb{R}^{d_{l-1}}$ and $c_0, \cdots, c_k$ are constants.

Because the filters only learn the approximated neighborhood $A_i^{(l)}$ (i.e. the *summation* of neighbors' node embeddings). The relative information of neighbors' neighborhoods is lost due to the summation.

(3) Task model: generate predictions

Given graph embedding $h^{(L)} \in \mathbb{R}^{N \times d_L}$ generated by Ego-CNN task models such as SVM or classification layers can be used to generate predictions for the learning task.

**Effective Receptive Field on Ambient Graph** A receptive field in Ego-CNN has the ambient coverage on graph that is effectively enlarged as network depth increases. Figure 6 shows the coverage of a node's receptive field across layers. Originally, this node's neighborhood is a 1-hop ego networks (Figure 6(a)). The coverage of its corresponding receptive field is expanded into a 2-hop ego network (Figure 6(b)) by passed through 1 Ego-Convolution. Note that it is not exactly its 2-hop ego network since edges across two neighbor's ego networks are not captured. As the depth of Ego-CNN increases, a receptive field at the $l$-th Ego-Convolution layer approximates covers a $(l+1)$-hop ego network. As shown in Figure 6(e), it's possible to cover the entire graph by a single receptive field.

The effectiveness of ambient coverage on graphs enables Ego-CNN to detect global-scale structures.

Besides, the effective coverage can be quite different for different receptive fields, which may reflect the positional information of the nodes. For example, the receptive field shown in Figure 6(e) can cover the entire graph, while the one in Figure 6(f) only covers a small part of the graph. This

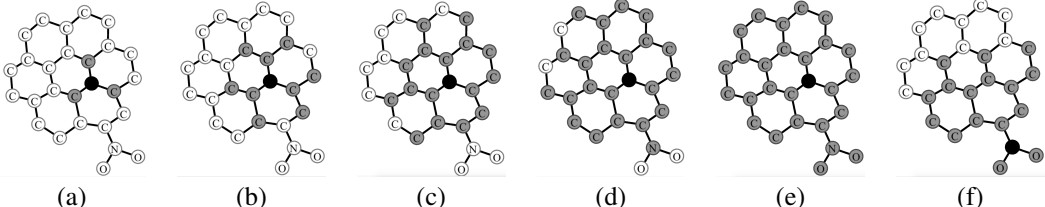

$$\text{(a)} \qquad \text{(b)} \qquad \text{(c)} \qquad \text{(d)} \qquad \text{(e)} \qquad \text{(f)}$$

Figure 6: A receptive field of Ego-CNN effective enlarges the coverage on graph as depth increases. (a) Suppose input neighborhoods are 1-hop ego networks. (b)-(e) the effective coverage of receptive fields at the 1st to the 4-th Ego-Convolution layer, which are the neighborhoods (in grey) centered at the same node (colored in black) at different scales. (f) the coverage of receptive field of another node at the 4th layer only covers a small local region of graph. The different coverage on graph reflects the position of the center node.

difference is related to the global position of these nodes. By comparing receptive fields, we think it is possible for Ego-CNN to be aware of the positional information of nodes and then take advantages of it.

## 4 ADVANTAGES OF EGO-CNN

### 4.1 DETECTING CRITICAL STRUCTURES

Given a trained Ego-CNN, there are many ways to visualize critical structures. Here we consider Ego-CNN is used with the general task model, which can be several neural network layers or can be an additional classifier such as SVM. Our approach takes in two steps.

1. Find out the most important neighborhoods.
   The simplest way is to add an Attention layer (Itti et al. (1998)) to generate a score representing the importance of each neighborhood. To train the weights of the Attention layer, we fix weights of previous layers (i.e. Patchy-San and Ego-Convolution) and add a Dense layer to train for the task. We can pickup important neighborhoods by setting a threshold.

2. Identify critical regions in those important neighborhoods.
   Since the critical parts is considered in the network's perspective, we take the approach similar to Deconvolution (Zeiler et al. (2011)). For simplicity, assume we use ReLU as the activation function. Given the node embedding of an important neighborhood (found by step 1) $h_i^{(l)} \in \mathbb{R}^{d_l}$ at the $l$-th layer, we can interpolate its receptive field (which contains its and its $k$ neighbors' neighborhoods at previous layer) by $\sum_{j=1}^{d_l} h_{ij}^{(l)} W_j^{(l)} = \begin{bmatrix} h_i^{(l-1)} \\ h_{n_1}^{(l-1)} \\ \vdots \\ h_{n_k}^{(l-1)} \end{bmatrix}$,

   where $n_1 \ldots n_k \in Nbr(i)$. Note that the interpolated values belonging to the same position at the $(l-1)$-th layer are summed up, which is the same as Deconvolution on traditional CNNs. Then, for each of the interpolated neighborhoods, we iteratively repeat the Deconvolution to interpolate their neighborhoods at the preceding layer until reaching the Patchy-San layer getting many neighborhoods in the form of adjacency matrices. These adjacency matrix are critical structures to the task, and the edge weight in adjacency matrix corresponds to the importance of an edge.

### 4.2 EFFICIENCY AND THE SCALE-FREE PRIOR

**Computation Complexity** Given a graph with $N$ nodes. To generate neighborhood embeddings at the $l$-th Ego-Convolution. It takes $O(N)$ to lookup embeddings of $k$ neighbors' neighborhoods and $O(Nk \log k)$ to concat neighborhoods according to neighbors' global order, and $O\left(N(k+1)d_{l-1}d_l\right)$ to perform convolution through each receptive fields (of size $(k+1) \times d_{l-1}$) by

$d_l$ filters. In total, for a Ego-CNN with $L$ Ego-Convolution layers, it takes $O\left(N(k+1)\sum_{l=1}^{L}d_{l-1}d_l\right)$ to embed the graph.

**Scale-Free Regularizer** Scale-free networks (Li et al. (2005)) are networks with self-similarity, which means the *same patterns* can be observed when zooming at different scales. Specifically, scale-free networks can be determined by RS box-covering(Kim et al. (2007)) if it satisfies $N_B(l_B) \sim l_B^{s_B}$, i.e. to cover the entire network by fixed-length boxes, the box length $l_B$ and the required number of boxes $N_B(l_B)$ follows a power-law. In practice, the power-law degree distribution is usually a common indicator for scale-free networks.

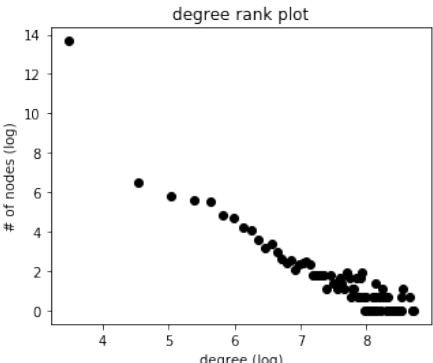

Figure 7: The degree distribution of Reddit dataset follows a power-law in log-log plot.

Figure 7 shows the degree distribution of the Reddit dataset, where we use it to benchmark model performance on classification tasks. We found degree distribution of graphs in Reddit datasets follows a power-law. This motivates us to think of the possibility of introducing scale-free priors into Ego-CNN. Recall that filters of the $l$-th Ego-Convolution layer is detecting the *combinations* of how neighborhoods at the $(l-1)$-th layer are combined into neighborhoods at the $l$-th layer. Because of the self-similarity in the definition, one way to generate the scale-free network is to repeat the combination at each layer. This means we can simply introduce scale-free prior into the Ego-CNN by weight-tying, i.e. let the filters share across different Ego-Convolution layers.

## 5 EXPERIMENTS

We perform a set of experiments to verify (i) the performance compared to existing graph embedding models on classification datasets, (ii) the effectiveness of scale-free regularizer, and (iii) visualization of identified critical structures. All experiments were performed on a computer with 48-core Intel(R) Xeon(R) E5-2690 CPU, 64 GB RAM, and NVidia Geforce GTX 1070 GPU. We use Tensorflow 1.0 to implement our methods.

### 5.1 GRAPH CLASSIFICATION

We benchmark on both bioinformatic and social-network datasets, which are processed by Kersting et al. (2016). Most graphs in bioinformatic datasets are labeled graphs that contain node and edge labels or attributes, while graphs are unlabeled (i.e. only structure of graph is given) in social network datasets. More details about these datasets can be found in DGK (Yanardag & Vishwanathan (2015)). The average test accuracy of 10-fold CV is reported to compared with existing methods. The results is listed in Table 2, where figures of most baselines are directly taken from their papers since experimental settings are the same. For simplicity, we use the same network architecture for all datasets. To embed neighborhood graphs, we choose Patchy-San since it deals with labeled graphs and works nicely with CNN visualization techniques. The architecture of network is composed of 1 Patchy-San layer (128 filter and k=10, which is the best setting for k reported in their paper), then follows our Ego-CNN with 5 Ego-Convolution layers (128 filters and k=16), and 2 Dense layers

with 128 neurons as the task model. To normalize neighbors, we follow Patchy-San's approach to use 1-WL labels. We train the network by Adam with learning rate 0.0001 and use Dropout with droprate 0.5 and Batch Normalization for regularization.

| Dataset | MUTAG | PTC | PROTEINS | NCI1 | DD |
|---|---|---|---|---|---|
| Size | 188 | 344 | 1113 | 4110 | 1178 |
| Max # node | 28 | 64 | 620 | 125 | 5748 |
| # class | 2 | 2 | 2 | 2 | 2 |
| WL kernel | 82.1 | 57.0 | 73.6 | 82.2 | 78.0 |
| DGK | 82.9 | 59.2 | 73.3 | 80.3 | |
| Subgraph2vec | 87.2 | 60.1 | 73.4 | 80.3 | |
| MLG | 84.2 | 63.6 | **76.1** | 80.8 | |
| Structure2vec | 88.3 | | | 83.7 | 82.2 |
| DCNN | 67.0 | 56.6 | | 62.6 | |
| Patchy-San | 92.6 | 60.0 | 75.9 | 78.6 | 78.1 |
| Ego-CNN | **93.1** | **63.8** | 73.8 | 80.7 | 75.6 |

| Dataset | IMDB (B) | IMDB (M) | REDDIT (B) | COLLAB |
|---|---|---|---|---|
| Size | 1000 | 1000 | 2000 | 5000 |
| Max # node | 270 | 176 | 3782 | 982 |
| # class | 2 | 3 | 2 | 3 |
| DGK | 67.0 | 44.6 | 78.0 | 73.0 |
| Patchy-San | 71.0 | 45.2 | 86.3 | 72.6 |
| Ego-CNN | **72.3** | **48.1** | **87.8** | **74.2** |

Table 2: 10-Fold CV test acc (%) on bioinfomatic(left) and social-network(right) datasets.

Our Ego-CNN improves the performance of Patchy-San on most datasets since the fixed-size neighborhoods (each contains at most 10 neighbors) detected by Patchy-San are grown into larger neighborhoods by Ego-Convolution. It makes sense on datasets such as MUTAG, where the task is to predict if compounds are mutagenic to DNA. It is reported by Debnath et al. (1991) that compounds with more than 3 benzene rings are very likely to be positive. To detect benzene rings in compounds, it's easier for Ego-CNN since we detect much larger local neighborhoods. This shows that representing graphs by larger neighborhoods is helpful on these datasets.

## 5.2 SCALE-FREE REGULARIZER

Since we found most graphs in Reddit dataset are scale-free graphs, we verify the effectiveness of scale-free regularizer on it. The results are shown in Table 3.

| Network architecture | Weight-tying? | 10-Fold CV test acc (%) | Total # of parameters |
|---|---|---|---|
| Patchy-San + 1 Ego-Convolution | | 84.9 | 1.3M |
| Patchy-San + 5 Ego-Convolution | ✓ | **88.4** | **1.3M** |
| Patchy-San + 5 Ego-Convolution | | 87.8 | 2.3M |

Table 3: Ego-CNN with scale-free prior on Reddit dataset.

To introduce scale-free priors into network by weight-tying, we expand 1 Ego-Convolution layer to 5 Ego-Convolution layers with the same weight, so the total parameters is 1.3M, which is the same as network in 1st row but the advantage is that we improve the test accuracy by 3.5%. Also, in the 3rd row, we compare against the network with same architecture but without using the scale-free regularizer. The number of parameter is 1M more but does not give better performance. This shows that introducing scale-free priors improves the learning efficiency, so that the same (or even better) performance can be obtained by training with fewer parameters.

## 5.3 VISUALIZATION OF CRITICAL STRUCTURES

As a sanity check, we generate two example compound datasets to visualize critical structures at the local-scale (Alkanes vs Alcohols) and global scale (Symmetric vs Asymmetric Isomers). The structures of compounds are generated of different size and under different vertex-orderings.

First, we'd like to know if the network considers OH-base as critical structures. The visualization result is shown in 8, where critical structures are plotted in grey color, and the node/edge size are proportional to its important score. We find that the OH-base in Alcohols is always consider critical no matter how long it its.

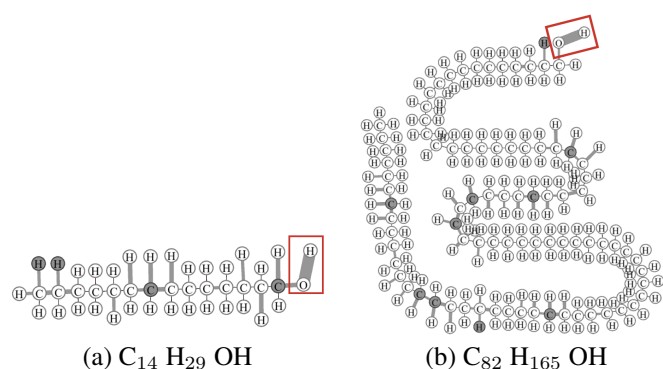

(a) $C_{14} H_{29} OH$        (b) $C_{82} H_{165} OH$

Figure 8: Visualization of critical structures on Alkane vs Alcohol. The critical structures are colored in grey and the node/edge size is proportional to its importance. The OH-base in Alcohols is always considered critical.

Next, we'd like to know how the network detects the concept of symmetry. The visualization result of Symmetric vs Asymmetric Isomers dataset is shown below. For symmetric isomers (Figure 9(a)), the critical structure, i.e. the symmetric hydrocarbon chains, is detected. By carefully observing the nodes and edge inside the detected structures (color in grey), we find an interesting thing that the importance of nodes/edges (i.e. which is plotted in different size proportional to the importance score for nodes and weights in interpolated adjacency for edges) are also roughly to be symmetric to the methyl-base. This symmetric phenomenon of detected structure can also be observed in asymmetric isomers (Figure 9(b)). We conjecture the network learns to compare if the two long hydrocarbon chains (which are branched from the methyl-base) are symmetric or not by starting comparing nodes and edges from the methyl-base all the way to the end of the hydrocarbon chains, which is similar to how people check if a structure is symmetric.

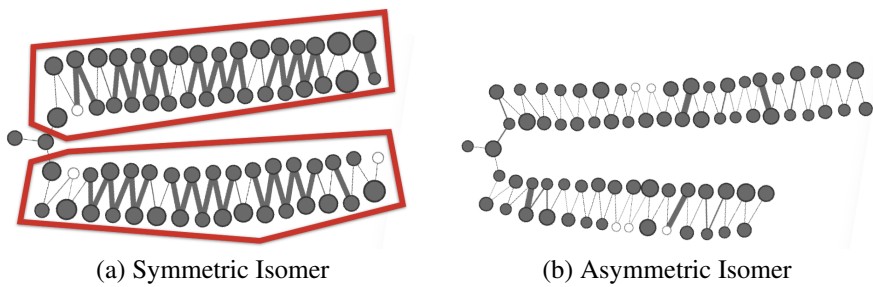

(a) Symmetric Isomer        (b) Asymmetric Isomer

Figure 9: Visualization of critical structures on Asymmetric Isomers vs Symmetric Isomers. The critical structures are colored in grey and the node/edge size is proportional to its importance. The critical patterns detected are roughly symmetric from the methyl branching node, which means the network somehow learned to count from the branching node to see if the structure is symmetric or not.

In the last experiment, we'd like to visualize the detected structures on Reddit dataset to see if it can help us to derive more knowledge about the dataset. In Reddit dataset, each graph represents a discussion thread. A node represent a user, and there's an edge if two users discuss with each other. The task is to classify the discussion style of threads. i.e. wether it is a discussion-based thread (e.g. threads under Atheism subreddit) or a QA-based thread (e.g. threads under AskReddit subreddit).

From the visualization result (Figure 10), we observe that the detected structures seemed to be the "bridges" that interconnect highly active users who has many discussions with other users. If each active user represents an strong opinion in the thread, the visualization result suggests that the discussion of different opinions are the key to discriminant discussion-based threads from QA-based threads.

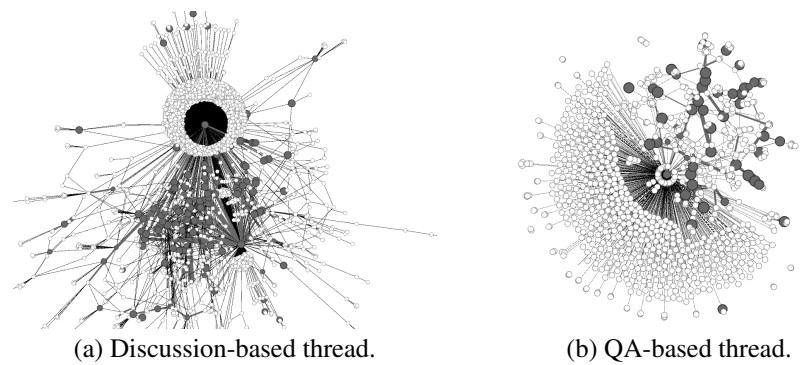

(a) Discussion-based thread.    (b) QA-based thread.

Figure 10: Visualization of critical structures on Reddit dataset. The critical structures are colored in grey and the node/edge size is proportional to its importance.

## 6 CONCLUSIONS

We propose Ego-CNN with idea of the egocentric design to define neighborhood on the same node across layers. The egocentric design enables Ego-CNN to fully take advantages of CNN to perform efficient learning on graphs, so that neighborhoods are enlarged as depth increases. Also, training efficiency can be increased by introducing scale-free priors, and visualization techniques for CNN can be used to visualize structures detected by Ego-CNN. We hope Ego-CNN helps people in domains such as bioinformatic, cheminformatics to understand their experiment results.

The future direction is to further reduce the space requirement of Ego-CNN. Since receptive fields at deeper Ego-Convolution layer represents larger neighborhoods, the overlapping area among neighborhoods of different nodes may also be enlarged. So, instead of using all neighborhoods, it might be enough to embed graphs by only part of the neighborhoods.

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
