# OpenReview forum: "Ego-CNN: An Ego Network-based Representation of Graphs Detecting Critical Structures"
_ICLR.cc/2018/Conference — Reject_

### Official Review · AnonReviewer3 · 2017-11-27
**A well desgined architechture for task driven graph embedding**

**Rating:** 7
**Confidence:** 3

**Review:**

The authors proposed a convolutional framework based on merging ego-networks. It combines graph embedding layers with task driven output layers, producing interpretable results for critical structure detection. While based on existing embedding methods such as Patchy-San, the contribution of ego-centric convolution and multi-layer architechture is novel and has a lot of potential in applications. The overall presentation of the draft is also of high quality. I recommend its publication at ICLR.

Here is a list of suggested changes to further improve the draft,

1. The two panels of Figure 1 seems redundant.

2. Figure 4 does not provide useful information, especially in terms of how overlapping neighborhoods are aggregated at deeper layers.

3. There seems to be a mistake in Figure 5 with the top neighborhood in white

4. The connection between weight-tying and scale-free structure needs better explanation. Are the authors trying to say that fractal processes generates power-law degree distributions?

5. The visualization of critical structures are very helpful. However, it might be better to look into structures in high level layers for truly global signatures. This is especially the case for the reddit dataset, where visualizations at the node and edge level creates hairballs.

---

> ### Author Response · Authors · 2017-12-19
> **Re: A well designed architecture for task driven graph embedding**
>
> Dear Reviewer,
> Thank you for the positive comments and useful suggestions. We have revise based on your suggestions.
>
> Q: >> “1. The two panels of Figure 1 seems redundant.”
> A: Thank you for the suggestion. Indeed, having Figure 1(b) is enough.
>
> Q: >> “2. Figure 4 does not provide useful information, especially ...”
> A: Thank you for pointing out. We have include that in Figure 4.
>
> Q: >> “3. There seems to be a mistake in Figure 5 with the top neighborhood in white”
> A: You are right. Thanks for pointing out.
>
> Q: >> “4. The connection between weight-tying and scale-free structure needs better explanation. Are the authors trying to say that fractal processes generates power-law degree distributions?”
> A: Thanks, we have cited the definition of a Scale-Free network [1] and added further explanations to the paper, as extracted below:  “Scale-Free networks [1] are networks with self-similarity, which means the same patterns can be observed when zooming at different scales.”
> The power-low distribution shown in the paper is a common indicator for scale-free networks.
> And, by the definition of “self-similar” property, the weight-tying (i.e.repeat the combination of neighborhood patterns at each layer) is a natural way to generate scale-free networks(, and the power-law degree distribution will surely follow).
>
> Q: >> 5. “it might be better to look into structures in high level layers for truly global signatures ...”
> A: Thank for your suggestion. Hopefully, we would try other way to show global signatures before 5 Jan.
>
> [1] LI, Lun, et al. “Towards a theory of scale-free graphs: Definition, properties, and implications”. Internet Mathematics, 2005.

---

### Official Review · AnonReviewer2 · 2017-11-28
**Interesting topic but poor/informal presentation**

**Rating:** 4
**Confidence:** 4

**Review:**

Dear authors,

Thank you for your contribution to ICLR. The problem you are addressing with your work is important. Your paper is well-motivated. Detecting and exploiting "critical structures" in graphs for graph classification is indeed something that is missing in previous work.

After the introduction you discuss some related work. While I really appreciate the effort you put into this section (including the figures etc.) there are several inaccuracies in the portrayal of existing methods. Especially the comparison to Patchy-san is somewhat vague. Please make sure that you clearly state the differences between patchy-san and Ego-CNNs. What exactly is it that Patchy cannot achieve that you can. I believe I understood what the advantages of the proposed method are but it took a while to get there. Just one example to show you what I mean; you write:

"The reason why the idea of Patchy-San fails to generalize into multiple layers is that its definition of
neighborhood, which is based on adjacency matrix, is not static and may not corresponding to local
regions in the graph. "

It is very difficult to understand what it is that you want to express with the above sentence. Its definition of neighborhood is based on adjacency matrix - what does that mean? A neighborhood is a set of nodes, no? Why is it that their definition of neighborhood might not correspond to local regions? In general, you should try to be more precise and concise when discussing related work.

Section 3, the most important section in the paper that describes the proposed Ego-CNN approach, should also be written more clearly. For instance, it would be good if you could define the notion of an "Ego-Convolution layer." You use that term without properly defining it and it is difficult to make sense of the approach without understanding it. Also, you contrast your approach with patchy and write that "Our main idea is to use the egocentric design, i.e. the neighborhood at next
layer is defined on the same node." Unfortunately, I find it difficult to understand what this means. In general, this section is very verbose and needs a lot more work. This is at the moment also the crucial shortcoming of the paper. You should spent more time on section 3 and formally and more didactically introduce your approach. In my opinion, without a substantial improvement of this section, the paper should not be accepted.

The experiments are standard and compare to numerous existing state of the art methods. The data sets are also rather standard. The one thing I would add to the results are the standard deviations. It is common to report those. Also, in the learning for graph structured data, the variance can be quite high and providing the stddev would at least indicate how significant the improvements are.

I also like the visualizations and the discussion of the critical structures found in some of the graphs.

Overall, I think this is an interesting paper that has a lot of potential. The problem, however, is that the presentation of the proposed approach is verbose and partially incomprehensible. What exactly is different to existing approaches? What exactly is the formal definition of the method? All of this is not well presented and, in my opinion, requires another round of editing and reviews.

---

> ### Author Response · Authors · 2017-12-19
> **Re: Interesting topic but poor/informal presentation**
>
> Dear reviewer,
> Thank you for time and constructive comments. Here are our answers to your questions.
>
> Q: What exactly Patchy-San cannot do but we can?
>
> Both of Patchy-San and ours can detect useful patterns, but ours are more efficient in terms of the size of detectable local regions.
>
> Remind that Patchy-San scans the k x k adjacency matrix of neighborhoods formed by the k nearest neighbors. However, Patchy-San (proposed in their paper) is only a “single layer” model, meaning that it can only detect local neighborhoods with at most k nodes. By contrast, the detectable size of local neighborhoods of our Ego-CNN is increased as depth increases.
>
> In Section 2, we tried to “generalize” Patchy-San’s idea (detecting patterns in the adjacency matrix of a node) to multiple layers and showed that generalization fails.
> However, Patchy-San cannot be directly stacked into multiple layers because the output of Patchy-San(i.e. neighborhood embeddings) cannot be treated as the required input "adjacency matrix" of the next Patchy-San layer.
> A naive way to generate the required k x k adjacency matrix is to calculate the pairwise similarity of the k neighborhoods with the most similar embeddings.
> This generalization does enlarge the receptive fields of Patchy-San as depth increases.
>
> However, as stated in Section 2, it would be very hard to realize in practice because of two reasons:
> (1) similar neighborhoods are selected based on the “output” of previous layer. During training, the output of previous layer is likely to change, making the composition of a neighborhood “non-static”.
> (2) similar neighborhoods may not be adjacent at a deeper layer, preventing the enlarged receptive field of a neuron from denoting a local neighborhood (i.e. connected subgraph) .
>
> The components forming into a neighborhood at deeper layer are likely to change and may spread out to the "entire graph" during training. Thus, this generalization (based on adjacency matrix) does not give neighborhoods corresponding to local regions. And our egocentric design is designed to solve the above problems.
>
> Back to the sentences that are confusing to you:
> >> “The reason why the idea of Patchy-San fails to generalize into multiple layers is that its
> >> definition of neighborhood, which is based on adjacency matrix, is not static and may not
> >> corresponding to local regions in the graph. ”
>
> Thank you for picking them out. We meant to briefly mention the drawbacks of the “generalized” version. In fact, the “adjacency matrix” is the one defined on the neighborhood embeddings.
> Here is an update with more details:
> “The reason why the idea of Patchy-San fails to generalize into multiple layers is that its definition of neighborhood (which is based on adjacency matrix) makes the composition of neighborhoods at deeper layer not static and may not corresponding to local regions in the graph. ”
>
> Q: >> "Our main idea is to use the egocentric design, i.e. ..." Unfortunately, I find it difficult to understand
>      >> what this means.
> A: Thanks for your constructive comments, we have revised Section 3 based on your suggestions.
>
> Q: What exactly is the formal definition of the method?
> A:We replace the Algorithm describing steps of Ego-Convolution with formal definition in math formula. Please refer to the Section 3.
>
> If you have further questions, please comment below. And we appreciate if you could update your rating if the above clears your doubts.

---

### Official Review · AnonReviewer4 · 2017-12-11

**Rating:** 4
**Confidence:** 4

**Review:**

The paper proposes a new method (Ego-CNN) to compute supervised embeddings of graphs based on the neighborhood structure of nodes. Using an approach similar to attention and deconvolution, the paper also aims to detect substructures in graphs that are important for a given supervised task.

Learning graph representations is an important task and fits well into ICLR. The paper pursues interesting ideas and shows promising experimental results. I've also found the focus of the paper on interpretability (by detecting important substructures) interesting and promising. However, in its current form, I am concerned about both the novelty and the clarity of the paper.

Regarding novelty: The general idea of Ego-CNN seems to be quite closely related to the model of Kipf and Welling [2]. Unfortunately, this connection is neither made clear in the discussion of related work, nor does the experimental evaluation include a comparison. In particular, the paper mentions that Ego-CNN is similar to the Weißfeiler-Lehman (WL) algorithm. However, the same is the case for [2] (see Appendix A in [2] for a discussion). It would therefore be important to discuss the benefits of Ego-CNN over [2] clearly, especially since [2] is arguably simpler and doesn't require a fixed-size neighborhood.

Regarding clarity: In general, the paper would greatly benefit from a clearer discussion of methods and results. For instance,
- The paper lacks a complete formal definition of the model.
- Detecting critical substructures is an explicit focus of the paper. However, Section 4.1 provides only a very short description of the proposed approach and lacks again any formal definition. Similarly, the experimental results in Section 5.3 require a deeper analysis of the detected substructures as the presented examples are mostly anecdotal. For instance, quantitative results on synthetic graphs (where the critical substructures are known) would improve this section.
- The discussion of scale-free regularization in Section 4.2 is very hand-wavy. It lacks again any formal proof that the proposed approach exploits scale-free structures or even a proper motivation why this regularization should improve results. Furthermore, the experimental results in Section 5.2 are only evaluated on a single dataset and it is difficult to say whether the improvement gains are due to some scale-free property of the model. For instance, the improvement could also just stem from the different architecture and/or decreased overfitting due to the decreased number of parameters from weight-tying.

Further comments:
- The discussion of related work is sometimes unclear. For instance, precisely why can't Neural Fingerprint detect critical structures? Similarly, how is the k-node neighborhood constraint of Patchy-San different than the one of Ego-CNN?
- In graph theory, the standard notion of neighborhood are all nodes adjacent to a given node, e.g., see [1]
- The writing could be improved, since I found some passages difficult to read due to typos and sentence structure.

[1] https://en.wikipedia.org/wiki/Neighbourhood_(graph_theory)
[2] Kipf et al. "Semi-supervised classification with graph convolutional", 2017.

---

> ### Author Response · Authors · 2017-12-19
> **Re: Review**
>
> Dear reviewer,
> Thank you for your time and comments. Before replying your specific comments, we think it is necessary to clarify some misunderstandings first.
>
> First, the model of Kipf and Welling [1] is not quite related to our paper, and this may have misled you from judging the novelty.
>
> You said:
> >> Regarding novelty: The general idea of Ego-CNN seems to be quite closely related to the model of Kipf and Welling.
>
> Our main idea is to detect the “precise” critical structure but not aiming to be a generalization of Weisfeiler-Lehman(WL).
> In fact, as you pointed out in Appendix A.1 of Kipf and Welling [1], to be a generalization of WL, it only requires the algorithm to approximate the “hash function” in WL. And to generalize WL, convolving on the “summation” of neighbors’ node embeddings is enough.
>
> However, being a generalization of WL is not enough to detect the “precise” neighborhood structure. As “summing” over neighbors’ node embeddings loses the relative position of neighbors’ neighborhoods (which is also the same problem as Neural Fingerprints [2] and is discussed in Section 2 in the draft).
>
> And that is the reason why we design our filters to learn the “entire” neighborhood structure, but not an approximation (i.e. “summation” of neighboring node embeddings). Although the math formula may look similar, the underlying ideas are quite different. It is the goal of detecting precise structure that separates us from those previous works such as Kipf and Welling[1], and Neural Fingerprints[2].
>
> We hope the above explanation can help you understand the fundamental difference between our work and Kipf and Welling [1].
>
> The following answers your specific comments:
>
> Q: The paper lacks a complete formal definition of the model
> A: Thank for your comment.
> We have  replaced the Algorithm describing Ego-Convolution with formal math definition in the revision.
> Also, more explanation is added to section 4.1 visualization based on your suggestion.
>
> Q: >> - The discussion of scale-free regularization in Section 4.2 is very hand-wavy...
> A: Thanks, we have cited the definition of a Scale-Free network [3] and added further explanations to the paper, as extracted below:  “Scale-Free networks [3] are networks with self-similarity, which means the same patterns can be observed when zooming at different scales.”
> The power-low distribution shown in the paper is a common indicator for scale-free networks.
> And, by the definition of “self-similar” property, the weight-tying (i.e.repeat the combination of neighborhood patterns at each layer) is a natural way to generate scale-free networks(, and the power-law degree distribution will surely follow).
>
> Q: why can't Neural Fingerprint detect critical structures?
> A: It is that the filters in Neural Fingerprint only learn the approximated neighborhood(i.e. summation of neighbors’ node embeddings). Neural Fingerprint is basically a hash function that maps a graph to a unique fingerprint. But you cannot do the inverse to derive precise structure from a given fingerprint in their model(due to the summation in their design) and they do not need to be able to.
> Although, you may argue it is possible to use a dictionary to store the mapping. But that additional dictionary is not even needed in our case. The above has been explained in Section 2.
>
> Q: Similarly, how is the k-node neighborhood constraint of Patchy-San different than the one of Ego-CNN?
> A: Sorry, we do not understand your question very well. Are you asking why the k used in Patchy-San layer and the one in Ego-Convolution can be different? (ex: k=10 in Patchy-San layer, and k=16 in Ego-Convolution)
> Because our main idea is to “aggregate” k neighbors’ neighborhoods, the size of the neighborhoods has nothing to do with the aggregation. As long as the neighborhoods are centered at the corresponding node, our Ego-Convolution can generate enlarged neighborhoods by aggregation.
>
> If you have further questions, please feel free to comment below. And we appreciate if you could update your rating if the above clears your doubts.
>
>
> [1] Kipf et al. "Semi-supervised classification with graph convolutional", ICLR 2017.
> [2] Duvenaud et al. “Convolutional networks on graphs for learning molecular fingerprints”, NIPS 2015.
> [3] LI, Lun, et al. “Towards a theory of scale-free graphs: Definition, properties, and implications. Internet Mathematics”, 2005.

---

### Author Response · Authors · 2017-10-29
**Correction of Typos**


* In the 4th paragraph of section 1., The only work ... is Spatial GCN ..., but it has the complexity O(N^2), ... should be O(N^3).
* In section 3. Effective Receptive Field on Ambient Graph, all reference to Figure 4.2 should be Figure 6.
* Table 4. caption should be "Ego-CNN with ..."
* Figure 8(b) caption should be C_82 H_165 OH

---

### Author Response · Authors · 2017-12-19
**Announcement for reviewers**

A revision updated (latest updated on 5 Jan.)
* Section 3
 - replace Algorithm steps of Ego-Convolution with formal definition
 - add comparison to previous work to show why they fail to detect precise structure
 - rewrite verbose sentences without changing the meaning and fix typos
* Section 4
 - add details to visualization steps
 - add definition of scale-free

---

### Decision · Program_Chairs · 2018-01-29
**ICLR 2018 Conference Acceptance Decision**

**Decision:**

Reject

**Comment:**

This paper deals with the important topic of learning better graph representations and shows promise in helping to detect critical substructures of graph that would help with the interpretability of representations. Unfortunately, this work fails to accurately portray how it relates to previous work (in particular, Niepert et al, Kipf et al, Duvenaud et al) and falls short of providing clear and convincing explanations of what it can do that these models can't, without including all of them in experimental comparisons.